# Application of Selected Nanomaterials and Ozone in Modern Clinical Dentistry

**DOI:** 10.3390/nano11020259

**Published:** 2021-01-20

**Authors:** Adam Lubojanski, Maciej Dobrzynski, Nicole Nowak, Justyna Rewak-Soroczynska, Klaudia Sztyler, Wojciech Zakrzewski, Wojciech Dobrzynski, Maria Szymonowicz, Zbigniew Rybak, Katarzyna Wiglusz, Rafal J. Wiglusz

**Affiliations:** 1Department of Experimental Surgery and Biomaterial Research, Wroclaw Medical University, Bujwida 44, 50-345 Wrocław, Poland; adam.lubojanski@student.umed.wroc.pl (A.L.); wojciech.zakrzewski@student.umed.wroc.pl (W.Z.); maria.szymonowicz@umed.wroc.pl (M.S.); zbigniew.rybak@umed.wroc.pl (Z.R.); 2Department of Pediatric Dentistry and Preclinical Dentistry, Wroclaw Medical University, Krakowska 26, 50-425 Wrocław, Poland; maciej.dobrzynski@umed.wroc.pl; 3Institute of Low Temperature and Structure Research, Polish Academy of Sciences, Okólna 2, 50-422 Wrocław, Poland; n.nowak@intibs.pl (N.N.); j.rewak@intibs.pl (J.R.-S.); 4Department of Pediatric Dentistry, Academic Dental Polyclinics, Krakowska 26, 50-435 Wroclaw, Poland; klaudia.sztyler1@gmail.com; 5Student Scientific Circle at the Department of Dental Materials, School of Medicine with the Division of Dentistry in Zabrze, Medical University of Silesia in Katowice, Akademicki Sq. 17, 41-902 Bytom, Poland; wojt.dobrzynski@wp.pl; 6Department of Analytical Chemistry, Wroclaw Medical University, Borowska 211 A, 50-566 Wroclaw, Poland; katarzyna.wiglusz@umed.wroc.pl

**Keywords:** dental nanomaterials, dental implants, endodontics, prosthetic, ozone, antimicrobial activity, microorganisms

## Abstract

This review is an attempt to summarize current research on ozone, titanium dioxide (TiO_2_), silver (Ag), copper oxide CuO and platinum (Pt) nanoparticles (NPs). These agents can be used in various fields of dentistry such as conservative dentistry, endodontic, prosthetic or dental surgery. Nanotechnology and ozone can facilitate the dentist’s work by providing antimicrobial properties to dental materials or ensuring a decontaminated work area. However, the high potential of these agents for use in medicine should be confirmed in further research due to possible side effects, especially in long duration of observation so that the best way to apply them can be obtained.

## 1. Introduction

Ozone is a gas composed of three oxygen atoms: the molecular weight is 47.98, it is an unstable substance, which quickly releases single oxygen atoms, and the half-life is 40 min at 20 °C and nearly 140 min at 0 °C. Ozone is colorless and has a characteristic smell. It occurs in nature as the ozone layer which protects organisms from ultraviolet rays. Ozone is heavier than air and it moves down where gas combines with pollutants, which is known as the self-cleansing phenomenon [1,2,3]. It is an extremely strong oxidizer, 1.5 times stronger than chloride when comparing their anti-microbial potential [4]. A gas mixture of 0.5 to 5% of ozone and 95 to 99.5% of oxygen is used in medicine. For almost a century, singlet (nascent) oxygen is known for its ability to inactivate bacteria, fungi, and viruses. Progress in medicine opens new applications for ozone therapy [5,6].

Silver particles have been demonstrated to be effective components in adhesives, implants, or prosthetic materials. They have an electron configuration of [Kr]4d^10^5s^1^, and it is now possible to produce silver nanoparticles with controlled morphology and size, as well as specific target functions and homogeneity [7,8]. Additionally, they can be successfully used in orthodontics [9] as multifunctional building blocks for dental materials [10], components for tissue conditioner, as well as act synergistically with several types of antibiotics [11,12]. Antifungal effects of silver particles against *Candida albicans* can be especially observed when the material is added to silicone-based liners or resins [13]. With regard to antibacterial activity, particles of silver influence the permeability of bacterial membrane, leading to its disruption. The material is responsible for the stimulation of oxidative stress resulting in thedestruction of cellular structures, such as DNA [14], lipids or proteins, eventually causing the destruction of the entire bacterial cell. Abroad range of aspects of the synthesis, as well as application and toxicology of silver particles, have been covered in recent reviews [15,16].The time needed to fully release Ag^+^ from a particle depends on dissolution processes which depend on pH, the concentration of the dissolved O_2_, surface coating or ionic strength of the medium [17].

Copper oxide (CuO) plays a significant role as a bactericidal and antifungal agent [18]. It is the simplest element of Cu compounds, that reveal a range of potential physical properties such as high-temperature superconductivity, spin dynamics or electron correlation effects [19], and is cheaper than silver oxide. It has high surface areas with uncommon crystalline structures [20,21] and additionally can improve fluid viscosity and enhance thermal conductivity. These characteristics make CuO a potentially useful energy-saving material. Its nanoparticles are successfullyused as additives in lubricants, metallic coatings or polymers [22]. The extremely high surface areas and atypical morphologies make CuO nanoparticles enhance the shear bond strength of adhesives, additionally influencing their antimicrobial characteristics, which allow them to dose-dependently inhibit, for example, *Escherichia coli* bacilli, but not *Salmonella* Typhimurium [23]. *Streptococcus mutans* cocci, on the other hand, are affected by CuO in a similar way as they are affected by particles of silver [24].According to other studies, CuO can decrease biofilm formation from 70% to up to 80% [25].

Titanium dioxide (TiO_2_) is a photocatalyst and is widely used as a self-disinfecting and self-cleaning material for surface coating in a variety of applications [26].Currently, titanium and its alloys are broadly used in dental implantology due to their excellent biocompatibility and good mechanical characteristics. Thanks to its properties, including nontoxicity and super-hydrophobicity, it has been applied in removing bacteria and harmful organic materials from both air and water, acting as a sterilizing agent in places such as medical centers [27]. However, TiO_2_ can be activated only in the low-ultraviolet (LUV)range (<400 nm). Apart from its disinfecting characteristics, recent research has proven that TiO_2_ allows localized drug delivery with the use of nanotubes [28].

Metal nanoparticles can be obtained by converting metals into fine particles with a diameter smaller than 100 nm. Platinum nanoparticles (PtNPs) have been reported in several studies [29,30] to have antibacterial and anti-inflammatory effects, and because PtNPs were demonstrated to be a potent antioxidant in vitro, their addition to resin-based materials may also improve their biocompatibility [31]. Additionally, platinum particles mixed with a 4-methacryloyloxyethyl trimellitic anhydride (4-META)/methyl methacrylate (MMA) adhesive increase the dentin bond strength twice when compared to the regular material [32]. Contact between Pt particles and bacteria has reportedly led to the decomposition of the latter [33]. Yang et al. [34] confirmed that platinum particles are unlikely to cause allergy, have a potential for clinical application and do not cause genotoxic potential.This review is focused on using nanotechnology and ozone in modern dentistry. Moreover, the antibacterial properties of the agents can allow for more effective methods of treatment with fewer complications.

## 2. Antimicrobial Properties

### 2.1. Gaseous Ozone, Ozonated Water and Ozonated Oil

In stomatology, three different forms of ozone are used: gaseous ozone, ozonated water and ozonated oil (sunflower oil, olive oil, groundnut oil) [35].

There are numerous research studies describing the antibacterial activity of ozone against oral pathogens. One of such studies evaluated the antibacterial effect of ozone against the pathogenic *Enterococcus faecalis* culture on a model of human teeth prepared using a specific protocol [36]. Samples were divided into four groups and for each of them, a different treatment was applied (ozone, photo-activated disinfection, saline as a positive control and NaOCl as a negative control). After 7 days of incubation, CFU (colony-forming unit) values were calculated. Both ozonation and photo-activated disinfection were revealed to be effective because the numbers of grown colonies were reduced [36]. Similar research was performed by Camacho-Alonso et al., who compared the activity of ozone with that of NaOCl solution, chlorhexidine, tri-antibiotic mixture, propolis and photodynamic therapy against *E. faecalis*. All the applied procedures were effective [37]. A comparison of a greater group of antibacterial procedures was conducted by Sancakli et al., who analyzed not only ozonation but also chlorhexidine and laser therapy Er:YAG (Erbium-doped Yttrium Aluminium Garnet laser), KTP crystal (PotassiumTitanylPhosphate crystal (KTiOPO_4_)), alone or in combination, against *Streptococcus mutans*. The obtained results revealed that the most efficient procedures were chlorhexidine, the combination of Er:YAG laser with ozone, Er:YAG with ozone and chlorhexidine (but its effect was weaker than for chlorhexidine alone) and the combination of KTP laser with ozone and ozone and chlorhexidine. Generally, ozonation and laser therapy applied alone were insufficient and required combining with other factors [38]. Besides gaseous ozone, also the antibacterial activity of ozonated olive oil was investigated using the direct contact agar diffusion test (the measurement of inhibition zones) on *Aggregatibacter actinomycetemcomitans*, *S. mutans* and *Prevotella intermedia* isolates. Ozonated olive oil inhibited the growth of all the tested strains but antibacterial effect was weaker than for chlorhexidine, which was used as a reference sample. The authors also evaluated minimal inhibitory concentrations (MIC) and minimal bactericidal concentrations (MBC) using the standard broth dilution method. Unfortunately, only a 10% solution of ozonated olive was able to inhibit the growth of *A. actinomycetencomitans,* and for *S. mutans* and *P. intermedia*, no MIC values could be determined using 0.01–10% solutions of the oil. Consequently, MBC values were also not determined. This result is not satisfactory, especially when compared with the formulation of chlorhexidine, where a 0.01% solution was already bactericidal [35]. There are also numerous research studies describing clinical trials with ozone. In the work of Krunić et al., ozone treatment was applied to patients with deep carious lesions. Chlorhexidine was applied in other patients as a reference. The experiments were performed on isolated *Lactobacillus* spp. as well as on the total population of bacteria. The data demonstrate that both applied treatmentprocedures caused a reduction in the number of bacteria, however, chlorhexidine was more effective [39]. In vivo experiments were also performed by Ajeti et al. The antibacterial efficacy of ozone combined with 0.9% NaCl, 2% chlorhexidine or 2.5% NaOCl was compared. The most effective of all of the tested combinations was the combination of ozone with NaOCl [40]. Apart from gaseous ozone and ozonated olive oil, ozonated water is also applied to reduce the risk of a microbial infection. In a clinical study by Anumula et al., patients were divided into two groups. One group received chlorhexidine, and the other ozonated water as an oral rinse. The saliva samples after 0, 7 and 14 days were diluted and cultivated to calculate CFU/mL values. The obtained results were very promising because the colony number of *S. mutans* decreased, and for ozonated water, the effect was even more visible than for chlorhexidine [41]. The findings of the research studies listed above confirm the antibacterial potential of different forms of ozone but, if possible, it should be combined with other techniques to obtain better results.

### 2.2. Titanium Dioxide Nanoparticles

Titanium oxide can reduce bacterial growth by damaging bacterial cell membranes, which leads to enhanced permeability and, as a consequence ofthe loss of vital cellular components, to death [42]. Antibacterial and antibiofilm effect of pure TiO_2_ coated on the surface of metal washers was investigated using the common strains inhabiting human oral cavity: *Streptococcus sanguinis*, *S. mutans* and *L. acidophilus*. In dentistry, biofilm contributes to formation of manyoral diseases (e.g. caries, periodontis, periimplantitis) therefore it is needed to ensure that necessary prevention and rapid reaction capabilities are in place to deal with any such problems. The simplified process of biofilm formation is presented in Figure 1.

The smallest reduction in biofilm formation was observed for *L. acidophilus,* but for *S. sanguinis* and *S. mutans,* the antibiofilm effect was more significant and dose-dependent—it increased with dopant enhancement [42]. A combination of TiO_2_ with metals is frequently described. In a work by Chen et al., the antibacterial activity against *S. mutans* was also investigated. TiO_2_ as well as Ag/TiO_2_ were mixed with polymethyl methacrylate (PMMA), silanized aluminum borate whiskers (ABWs) and nano-ZrO_2_ to obtain a composite. An additive of pure titanium oxide and its silver-modified derivative to the resin decreased bacterial growth without increasing the cytotoxicity of the material. Nevertheless, it should be noted that the sample with incorporated silver was much more effective [43]. Titanium dioxide, alone and combined with silver, was also combined with polyacrylate resin polymer to obtain nanohybrid coatings on titanium discs. These materials have been tested against *Streptococcus salivarius* and the obtained results are promising [44]. The antibacterial effect of TiO_2_ combined with silver was also investigated on *S. sanguinis* along with evaluating bacterial adhesion to the material in the form of titanium discs coated with TiO_2_ or Ag (alone or combined with hydroxyapatite). The results indicate that the discs coated with silver combined with hydroxyapatite were the most effective of all the tested samples [45]. In the work of Lavaee et al., *S. mutans* and *S. sanguinis* were isolated from saliva samples of patients and antimicrobial activity of TiO_2_, alone and combined with nano-Ag or nano-Fe_3_O_4_, was tested. The combination of these three nanoparticles (nano-TiO_2_ + nano-Ag + nano-Fe_3_O_4_) turned out to be the most effective of all the tested materials and this effect was observed for both tested strains. Moreover, the antibiofilm activity of this mixture was also observed [46]. Apart from silver, copper is another material that can be combined with titanium oxide to enhance its antimicrobial effect [47]. Titanium dioxide is also a promising material due to its photocatalytic properties: after UV exposure, it produces ROS, which enhances its antibacterial properties [48].

### 2.3. Silver Nanoparticles and Ions

Silver, especially in its ionic form, has the highest antimicrobial activity among metals. Its mode of action is complex and involves protein impairment by forming Ag-S bonds, which leads to respiratory chain dysfunction and damage to membrane pumps. Ag^+^ionsare also genotoxic—they interact with DNA which causes its condensation and inhibition of the replication process. Silver also stimulates ROS production, which is characteristic of metals [49]. Besides releasing ions, silver nanoparticles can also affect bacterial cells by themselves. One mode of action is based on attaching to the surface of a cell and leading to the denaturation of cell membrane as a result ofthe accumulation of particles. Due to their nano-size, they can also penetrate through the cell wall, and disturb metabolic processes and bacterial signal transduction [50]. There is an extensive body of research describing antibacterial activity of silver-based nanomaterials against oral pathogens. In the work of Liu et al., polyetheretherketone (PEEK) was coated with 3–12 nm of nano-silver coatings. The antimicrobial potential of the obtained materials was evaluated usingan *S. mutans* strain. The obtained results indicate that efficacy increases with the thickness of the silver layer, however even a 3 nm-thick sample effectively prevented bacterial growth. The number of bacterial colonies attached to the surface of silver-coated materials was also reduced [51]. *S. mutans* was also used in the research of Kim et al., who evaluated the antimicrobial potential of feldspathic porcelain combined with nano-sized silver at concentrations of 0%, 5%, 10%, 20% and 30%. All the tested samples, even the non-doped one, inhibited bacterial growth, but the most significant reduction was observed for the highest concentration of silver (30%) [52]. In order to evaluate antibacterial activity of silver nanoparticles, a model of human teeth (after extraction) was prepared and, after cleansing, contaminated with *E. faecalis*. After 21 days of incubation, the samples were divided into three groups treated with different intracanal dressings: Ca(OH)_2_ paste, Ca(OH)_2_ paste mixed with chlorhexidine and Ca(OH)_2_ paste supplemented with a suspension of silver nanoparticles. Samples prepared in such a way were incubated for 1 week and 1 month and then the colonies were counted. Moreover, SEM (Scannicelectron Microscopy) images were measured. The obtained data revealed that all of the applied dressings effectively reduced bacterial growth and adhesion, and the silver additive improved this effect [53]. Interesting results were also obtained by Yin et al., who prepared sodium fluoride solution mixed with different concentrations of polyethylene glycol-coated silver nanoparticles (PEG-AgNPs). Antibacterial activity (half maximal inhibitory concentration, IC_50_) of these materials was examined using an *S. mutans* strain. The obtained data indicate that PEG-AgNPs effectively kill bacteria at a concentration of 21.16 ppm ofsilver and this concentration was non-cytotoxic for a human gingival fibroblast cell line (HGF-1) [54]. In the work by Bacali et al., antibacterial activity of polymethyl methacrylate (PMMA) denture resin combined with graphene and Ag nanoparticles was tested against, among others, *S. mutans*. Interestingly, pure PMMA also inhibited *S. mutans* growth but the addition of graphene and silver nanoparticles intensified this effect [55]. Three oral microbes (*E. faecalis*, *S. mutans* and *Streptococcus oralis*) were used to evaluate the antimicrobial activity of bioactive glass combined with silver nanoparticles and tetracycline. The prepared materials effectively released the loaded drug, which led to the inhibition of bacterial growth. However, materials without the addition of tetracycline were not so effective. Silver-combined bio-glass slightly inhibited the growth of *E. faecalis* but no such effect was observed for *S. mutans* [56]. Chitosan was also used as a carrier of drug (gentamicin) and silver nanoparticles. Pure chitosan slightly reduced bacterial growth (*S. mutans*) but adding gentamicin or the combination of Ag/chitosan or Ag/chitosan/gentamicin enhanced the antibacterial effect. Surprisingly, no differences between the action of these three materials were detected [57]. Epigallocatechin gallate was another material used as a carrier for nanoparticles. Antimicrobial efficacy was evaluated for silver nanoparticles and compared with ionic silver in the form of AgNO_3_. The results indicate that the effectiveness of silver nanoparticles against *S. mutans* is higher than that of the AgNO_3_solution. The red-to-green ratio calculated from confocal laser microscopy (dead/live cells proportion) was also higher for AgNPs. Moreover, in microscopic images, it can be seen that bacteria did not form a biofilm structure in the AgNPs sample, whilst for AgNO_3_ and control (water), a fully grown biofilm can be noted. Additionally, AgNPs significantly reduced the production of lactic acid and polysaccharides by *S. mutans* biofilms [58].

### 2.4. Platinum Nanoparticles

The antibacterial effect of platinum is based on the ability to decompose bacterial cell structure. The exact mechanism is uncertain and, as for metals in general, probably connected with the increase of reactive oxygen species (ROS) production, which results in cellular damage via oxidative stress [33]. Platinum nanoparticles (PtNPs) have an antimicrobial effect against oral pathogens. In the work of Itohiya et al., PtNPs were obtained via direct irradiation of platinum with an infrared pulsed laser and tested on common dental bacteria: *S.mutans*, *E. faecalis* and *Porphyromonasgingivalis*. The range of applied concentrations of PtNPs was 1–20 ppm and the concentrations above 5 ppm completely inhibited the growth of all the tested strains [33]. Platinum nanoparticles were also combined with polymeric PMMA and the antibacterial activity of such a combination was evaluated using *S. mutans* and *Streptococcus sobrinus* reference strains. The results indicate that platinum nanoparticles exhibit antimicrobial activity against the planktonic form of both tested strains (growth reduction). Moreover, platinum-combined polymer turned out to be less prone to bacterial adhesion than pure PMMA [59].

### 2.5. Copper Oxide

Copper oxide nanoparticles (CuONPs) can be obtainedwith the use of various methods of synthesis. By using different preparation methods, particles with different sizes can be produced; for example, through colloidal-thermal synthesis, precipitation synthesis, microwave irradiation or sol-gel techniques, very small particles can be obtained which are estimated from 3 to 10 nm. Larger particles from 10 to even 30 nm can be received via sonochemical synthesis, spinning disk reactor, solid-state reaction, microemulsion system or thermal decomposition [60].

CuONPs, due to their small size, are well known for their optical and magnetic properties and electrical conductivity. However, copper oxide nanoparticles are also known for their biological properties, in particular for their antimicrobial activity; nevertheless, cytotoxic properties were well estimated in both in vitro and in vivo models [60].

Skin and the respiratory system are parts of the human body which are the most exposed to CuOPNs. The cytotoxicity of copper oxide nanoparticles against normal cell lines such as human keratinocytes (HaCaT)and mouse embryonic fibroblasts (MEF) was estimated by Luo et al. Copper oxide nanoparticles, in a dose-dependent manner (from a concentration of 20 µg/mL), distinctly induced c-Jun N-terminal kinase (JNK) and extracellular signal-regulated kinase (Erk); moreover, level of p53 was significantly decreased in HaCaT and MEF cell lines, and a reduction of the amounts of viable cells was also observed in a concentration-dependent way, which clearly indicates the toxic effect on the above-mentioned cell types [61]. Another study confirmed the cytotoxic effect of CuONPs on human lung epithelial cells (A549). Concentration from 10mg/mL to 100µg/mL drastically decreased cell viability. CuONPs particles were compared with CuO bulk particles and it was clearly demonstrated that CuO bulk particles exhibitedaweaker toxic effect (concentration 58 mg/mL) than CuO nanoparticles (concentration of 15 mg/mL). TEM images (Transmission Electron Microscopy) show the entry of CuOPNs into the intracellular environment, but also into the nucleus, mitochondria and lysosomes. Mitochondrial influx of CuO nanoparticles may lead to inducing its depolarization and probably caused the generation of reactive oxygen species (ROS) as well, which elevated oxidative stress [62]. The cytotoxic effect of CuONPson the A549 cell line was also confirmed by an independent research of Akhtar et al. [63]. The toxic effect of CuO nanoparticles against human blood lymphocytes was also evaluated by estimating cell viability, ROS generation, peroxidation of lipids, but also by estimating lysosomal and mitochondrial disruption. The data clearly indicated the disintegration of mitochondrial membrane and an elevated generation of reactive oxygen species (ROS), and lysosomal damage was also visible. The viability of human blood lymphocytes drastically dropped and was estimated at IC_50_ after the treatment with CuO nanoparticles at a concentration of 385 µM [64]. Sun et al. compared the cytotoxic properties of several different metal nanoparticles, such as Fe_3_O_4_, Fe_2_O_3_, SiO_2_, TiO_2_ and CuO, and exposed to them a human type II alveolar epithelial cell line (A549), human non-small cell lung cancer (H1650) cells and human nasopharyngeal carcinoma (CNE-2Z). Copper nanoparticles were determined to be the most toxic among all the tested nanoparticles and on all the tested cell lines, and the toxic effect was estimated at a concentration of 30 µg/mL [65]. Additionally, Beltrán-Partida et al. estimated the potential cytotoxic effect of CuNPs, CuCO_3_ and the antifungal drug triclosan on primary human gingival fibroblasts (HGF) isolated from a clinically healthy young (15 years old) male patient as a suitable model for this type of test. Among all the tested compounds, CuCO_3_ was the most toxic on HGF when compared to CuNPs which exhibited IC_50_ at a concentration of 137.4 μg/mL [66].

The antimicrobial potential of copper has been known for ages but, as a result ofthe development of nanotechnology, nano-sized copper oxide also became an issue of interest. The antibacterial effect of CuO was described by Ramazanzadeh et al., who prepared brackets coated with CuO, ZnO or a 1:1 combination of the materials which was tested on *S. mutans* after 0, 2, 4, 6 and 24 h. No effect was observed for ZnO but CuO completely inhibited bacterial growth after 4 h of incubation. The ZnO/CuO mixture was even more effective because the bacterial growth was affected after 2 h of treatment [67]. The susceptibility of *S. mutans* was also tested by Toodehzaeim et al., who prepared nanocomposites doped with 0.01%, 0.5% and 1% of CuO and the inhibition zones were measured around discs placed on the surface of agar plates with bacteria [68]. MIC_50_ values were determined for CuO by Amiri et al. on the *S. mutans*, *Lactobacillus acidophilus* and *Lactobacillus casei* strains. The obtained values are as follows: <1 µg/mL for *L. acidophilus*, 1–10 µg/mL for *S. mutans* and 10 µg/mL for *L. casei*. At the concentration of 1000 µg/mL, a complete reduction of growth was observed for all the tested bacteria [21].

## 3. The Use of Ozone and Nanoparticles in Dentistry

### 3.1. Restorative Dentistry

Tooth decay is a very common disease: The Global Burden of Disease Study 2017 reported that there are 2.4 billion people suffering from this illness, withcaries of primary teeth as a problem concerning 530 million children [69]. Ozonation is an efficient treatment that can help to reduce the amounts of microorganisms in oral cavity preventing to dental caries—the most common disease worldwide. In Figure 2 has been shown the number of oral diseases worldwide indicateing a serious problem for many people.

The consequence of primary tooth caries is a high risk of infection in permanent teeth; for this reason, it is important to prevent early childhood caries (ECC). Ozone therapy seems to be an appreciable method to supervise carious lesions. Ximenes, Cardoso, and Astorga et al. examined the potential of gaseous and aqueous ozone in the reduction of *S. mutans*, *L. acidophilus* and *E. Faecalis*: scientists took note of the last-mentioned bacteria, which is not a typical pathogen for ECC, but it is the species with the highest resistance to antimicrobial factors. Researchers point out good effectiveness of both methods in the evaluation of ECC [70]. Other studies show theefficacy of gaseous ozone in the reduction of *Lactobacillus* sp. in deep carious lesions to be similar to the efficacy of chlorhexidine [39]. On the other hand, Hauser-Gerspach et al. point out in their research that there is no significant difference in the reduction of viable bacteria in cavitated carious lesions in children in vivo, while aqueous ozone seems to be more effective in some cases [71]. Duangthip et al. also indicate a lack of efficacy of ozone in controlling dentin caries [72]. Another study points to ozone therapy as a good alternative for patients experiencing anxiety [6]. Kalnina [73] found no significant difference in the efficacy of ozone and fluoride varnish and sealants in reducing the probability of caries in permanent tooth. Similar study results were observed in the remineralization of initial caries in enamel. Due to these properties, ozone can be a powerful alternative for the prevention and treatment of caries. The author drew attention to the high cost of ozone generator compared to fluoride varnish and sealants [73]. Microleakage is a common problem in dentistry, withsecondary caries as its frequently observed effect. According to the study, ozone has no negative influence on pit and fissure sealants which protect the surface in more than 85% of caries in children. Ozone can be used to disinfect a previously prepared surface without affecting the adhesion of pit and fissure sealants [74]. A similar observation on the subject of initial micro-tensile bond strength of a self-etch adhesive to dentin was provided by Dalkilic et al., whoindicated a safe use of ozone as a disinfectant [75]. Another advantage of ozone is the absence of pain during the therapy, which makes it easier to work with anxious patients [76]. Almaz et al. noted the wide range of applications of ozone and effective properties in caries treatment, prevention, and remineralization. However, some researchers indicate inefficacy of ozone [77].

Mineral Trioxide Aggregate (MTA) is often used in the case of operator errors, such as root perforation or exposure of the pulp. This material can be modified with TiO_2_ and other nanoparticles to improve the properties. TiO_2_ provides an antimicrobial effect, self-cleaning and photo-elastic properties [78]. Another material commonly used in dentistry is glass ionomer (GIO), whose main components are calcium or strontium alumino-fluoro-silicate glass powder, which reacts with water-soluble acidic polymer. The resultant material is valued for the ability to release fluoride ions to the surface which stop the cariogenic process. Addition of TiO_2_ nanoparticles increases the durability of GIO (especially to compressive strength), which is a significant disadvantage of the material. Garcia-Contreras et al. [79] mentioned a better antibacterial effect of GIO modified with TiO_2_ nanoparticles. However, the sensitivity of GIO to water has increased, which caused the loss of material. Ferrando-Magraner et al. concluded in their scientific research that the most beneficial application of titania nanoparticles is for dental bonding material [80].

Cavities filled with dental material after inaccurate preparation can cause secondary caries. Theaddition of nanoparticles to materials can prevent the need of replacement. Composite resin modified with Ag doped with ZnO nanoparticles demonstrates antibacterial properties. Studies indicate that compressive strength does not change significantly in comparison to unenriched composite resin. Another property of nanoparticles is the reduction of bacterial biofilm, which is more resistant than planktonic bacterial cells [81], [82]. Koohpeima et al. used Ag nanoparticles as a coating before the application of etch-and-rinse and self-etch adhesive systems. The strength of bond did not change negatively, Ag NPs did not affect the color, which could have been the case under the influence of metal NPs. Together with Ag NPs, it increases the chances of proper treatment [83]. Vazquez-Garcia et al. indicates that calcium silicate cements, white MTA and Portland cement with 30% of ZrO_2_, after manipulation with AgNPs solution, have good effectiveness against *E. faecalis* in both forms of biofilm and planktonic cells. Both materials have similar application, and addition of NPs does not negatively affect their properties [84]. Elgamily et al., in studies concerning a cavity disinfectant containing AgNPs and AuNPs in comparison to chlorhexidine (CHX), examined their impact on *S. mutans*. Both agents reduced the amounts of bacteria, but CHX was more effective [85]. Silver diamine fluoride (SDF) is an effective anticariotic agent which currently includes AgNPsas a result ofthe development of nanotechnology. Scientists recommend it, especially for children, as the agent has good antimicrobial properties [86]. On the other hand, Fakhruddin et al. [87] point out that even though in vitro results are positive, there is no evidence of good effectiveness in in vivo studies.

Amiri et al. examined the influence of CuO Np on oral bacteria and *Candida* species. According to research, CuO NPs have good antibacterial properties but they have a weak impact on *Candida* species. Due to antibiotic resistance of bacteria, this nanoparticle may be a good control measure for preventing caries [21].

### 3.2. Endodontics

Endodontic treatment consists of proper canal preparation followed by hermetic filling. It can be achieved with the use of chemo-mechanical procedures which include abundant irrigation and effective canal shaping. Root canal instruments widen the main root canal space and remove most of the canal content, at the same time, mechanical interference produces smear layer which forecloses proper disinfection and prevents tight sealing of the root canal system. Chemical cleansing removes the smear layer and plays a major role in the overall success of endodontic treatment. The main reason why chemical interference and disinfection protocols are so crucial is the complexity of the root canal system and rich network of channels which are unreachable to instruments. The most important characteristics of irrigators are their tissue-dissolving ability and low toxicity, combined with antimicrobial effect [88]. The goal of modern endodontic therapy and modern endodontic materials is to achieve maximal disinfection of the root canal system, smallest possible area of preparation which will provide long-term success and no reinfections.Ozone is considered as a beneficial choice of canal antiseptic. Its main advantages are high antimicrobial activity, low toxicity, and the fact that they do not generate drug resistance [89]. In root canal treatment, ozone is useful to eliminate *Enteroccocus fecalis*, *Peptostreptococcus micros*, *Pseudomonas aeruginosa* and *Candida albicans*. Ozone is used in many forms such as gas, water, and oil. The water and gas forms can be used in the rinsing protocol. The oil form can be used as a medical insert in cases of dental pulp necrosis. The gas form provides high penetrability to lateral channels and root deltas which increases the chance of maximal disinfection. The concept of the device is simple, ozone is generated by the machine, then it is channeled through the handpiece to the affected root canals [90]. In Figure 3 has been shown an application of ozone therapy unit. Ozone is generated in the unit and then delivered to the root canal with an endodontics cannula. Silicone cap prevents ozone leakage. It has been shown deep penetration of ozone providing maximum disinfection rate.

After treatment, the gas is converted back into oxygen by the ozone neutralizer. According to the studies, the suggested method of ozone treatment in the root canal system is at the end of chemo-mechanical preparation. To improve its effect, the amount of organic debris left inside the root canal should be reduced to a minimum [91]. Many studies were conducted to verify assumptions connected with disinfection strength of ozone. Some of them noted that ozone water efficiency was similar tothat of 5.25% NaOCl [91,92,93,94,95,96], some of them obtained the same results with ozone gas, and in their case, the efficiency was similar to that of 3% NaOCl (Sodium hypochlorite) [97]. Results to the contrary were reported in some studies, in which after using ozone water irrigation, the canal content contained bacteria [98,99]. Ozone was investigated in many disinfection protocol combinations and according to a study of Hubbezoglu et al., the best action was accomplished when ozone was used at a concentration of 16 ppm with ultrasonic agitation [94]. Noites et al. achieved the best bacteria reduction effect with 2% chlorhexidine followed by 24 s of ozone [100]. This combination can be considered in the treatment of a resorbed apex, wide open foramen. Nevertheless, these indications should be estimated with caution [101]. Recaizan et al. compared the reduction of bacteria with the use of ozone to their reduction using other antiseptics. Bacteria reduction using ozone was 90.4% and it turned out to be more efficient than using Er.YAG and KTP lasers [99]. Scanning electron microscope examination was used in order tocomparedifferent treatment protocol results including a control group, NaOCl + EDTA (Ethylenediaminetetraacetic acid), NaOCl + citric acid, and NaOCl + ozonated water groups. Indentation distance increase turned out to be comparable in every protocol. Tubule densities, tubule diameters, total surface free energy and the Lewis base polar force were higher using NaOCl + EDTA treatment compared to the control group, NaOCl + citric acid and NaOCl + ozonated water treatments. The comparison of contact angle and wettability of three sealer materials shows that the use of NaOCl + ozonated water groups significantly increases the wettability of Acroseal on the root canal dentine and decreases contact angle compared to the control group. The results for Apexite and Endomethasone material yield distinctive results, in which the usage of NaOCl + ozonated water increases the contact angle compared to the control group [102]. The Study of Bojar et al. represented results in which AH- 26 and EX fill root canal sealers shows an improved shear bond after root canal ozone treatment [103]. The conclusion is that ozone has a certain antibacterial effect, its major advantage is non-toxicity and penetrability, however ozone cannot be used alone as acanal antiseptic, although studies show that it might improve root canal therapy and should be included in the treatment protocol.

Nanotechnology can also be applied in endodontic treatment. It is considered to be a stimulating and up-to-date topic that needs to be elaborated. Nanoparticles are minute solid particles with a diameter of 1–100 nm [104]. Their advantages are ultra-small sizes, high surface area to mass ratio and high chemical reactivity. Their antimicrobial effect is the result of large surface area and high charge density. AgNPs is one of the nanomaterials that can be considered in endodontic treatment. It has antibacterial effect on both Gram-positive and Gram-negative bacteria. It also shows antiviral and antifungal action. The mechanism responsible for the antimicrobial effect is still not fully understood. The most plausible mechanism is the interruption of ATP particle and preventing DNA replication by free silver ions, production of reactive oxygen or direct damage of cell membrane by Ag^+^. It is certain that free silver ions play a crucial role in antibacterial action. Scientific evidence shows that release of Ag^+^ from smaller particles (<10 nm) is higher than from largerones. One of the disadvantages of the AgNPs is possible toxicity. AgNPs above 80 μg/mL could be considered cytotoxic [24].Preclinical studies on rats show effectssuch as accumulation in organs, discoloration, cytotoxicity or even sperm cell production disturbance [105,106].

AgNPs can be used for coating gutta-percha. Shantiaee et al. compared nanosilver-coated gutta-percha to an uncoated one and it resulted in slightly less leakage [107]. Lotfi et al. compared AgNPs solution to NaOCl. The size of silver particle used in this examination was 35 nm. Their results report a similar antibacterial effect of AgNPs and 5.25% NaOCl [108]. González-Luna et al. obtained the same results but with comparison to 2.25% NaOCl [109]. A study of Abbaszadegan et al. shows satisfactory annihilation of *E. faecalis* [110]. Nevertheless, some studies show that AgNPs were less effective against *E. faecalis* than NaOCl, therefore, at the moment, AgNPs can not be considered as a substitute for NaOCl [104]. In Figure 4 has been shown, in simplification, the use of Ag^+^ ions in irrigation solution for endodontic application. Their advantages are ultra-small size, high chemical reactivity, and antimicrobial properties. The solution enables good disinfectant penetration and good antibacterial effect.

Another interesting application of AgNPs is adding them to the mineral trioxide aggregate (MTA). The idea was to improve activity against strict anaerobes. The outcome showed efficiency against different bacteria and *C. albicans* higher than in the case of unmodified MTA [111]. Another tested combination was Ca(OH)_2_ and AgNPs. The result was that Ca(OH)_2_ showed better antibacterial effect in comparison to AgNPs alone or AgNPs + Ca(OH)_2_ [109]. The addition of 0.15% AgNPs and 2.5% dimethylaminohexadecyl methacrylate (DMAHDM) to the AH Plus paste significantly increased antibacterial activity against *E. Faecalis* [112]. Chlorhexidine (CHX)-AgNPs containing lyotropic liquid crystals showed excellent sterilization and inhibitory effect on *E. faecalis* lasting for more than one month with a bacterial annihilation rate of 98.5%.

Nano-MgO can also be included in endodontic treatment. Its advantages are antimicrobial activity and non-toxicity. In one of the studies, nano-MgO was compared to different disinfectants such as CHX and NaOCl. There was no substantial difference between nano-MgO solutions, 5.25% NaOCl and 2% CHX gluconate regarding the time required to inhibit *E. faecalis* and *S. aureus* growth. On the other hand, 5 mg/L nano-MgO presented better overall antibacterial efficacy in the elimination of *E. faecalis* than 5.25% NaOCl [113]. Copper and copper oxide nanoparticles present both antibacterial and antifungal effect. They produce superoxide ions, reactive oxygen and cause leakage of intracellular components that causes cell death. Antibacterial effect may be provided by hydroxyl radicals, damage of DNA and important proteins [114]. A crucial stage to obtain desirable antibacterial effect is the synthesis of NPs followed by stability in the medium, which is usually a polymeric matrix. A study by Hajipour et al. shows that the antibacterial strength of copper nanoparticles depends on both nanoparticle and bacterial concentration, temperature, pH and aeration [115]. Except for *S. aureus*, CuO NPs show higher strength compared to other nanoparticles such as Sb_2_O_3_, ZnO or NiO. The role of CuO nanoparticles is to create an enriched surface of the inorganic antibacterial agents. Copper NPs show bactericidal effect similar to triclosan and decrease the level of many bacteria populations, such as *S. aureus*, *E. coli* and *B. subtilis*. Cu NPs show potential as a component in endodontic materials, however further studies are needed [116]. A layer of nanometric TiO_2_ can be used as a coating on super-elastic rotary NiTi instruments. The idea is to improve mechanical and chemical properties without losing cutting effectiveness. The study showed that TiO_2_ coating improved cutting capacity, corrosion behavior and fatigue resistance [117].

### 3.3. Dental Surgery

Ozone has a wide range of applications in treatments such as implant placement, hemisections, extractions, bone regeneration and tissue healing. Often, it depends on a patient’s condition, in these therapies’ antibiotics are used prophylactically. It isavery important factor in dentistry to ensure that the presence of microbes in the field of work is reduced to a minimum. Researchshow significantly faster regeneration, reduction of complications and pain. Even in difficult cases of patients with diabetes and microangiopathy, ozone therapy after topical application improves treatment. Animal research show better osseointegration in cases of reduced immunity [118]. Peri-implantitis is a frequently occurring inflammatory process caused by biofilm, it leads to bone loss and other complications [119]. In some research [120], scientists reported positive influence on the clinical attachment loss (CAL) and defect bone fill, which points to ozone decontamination as an effective method in bone healing in reconstructive surgery. Third molar surgery can cause pain, swelling and trismus.Kazancioglu et al. examined the influence of ozone. Due to its antimicrobial and analgetic properties, reduction of pain was observed when the gas was used one week before and one week after the extraction. However, ozone had no impact on trismus and swelling [121]. Other studies compared ozone and laser therapies: ozone was effective in pain treatment, as opposed to low-level laser therapies (LLLT), and it was also effective in reducing swelling and trismus. These methods can be alternative to corticosteroids and nonsteroidal anti-inflammatory drugs, which can cause side effects [122].

Titanium is commonly used in implantology owing to its adequate biocompatibility, however daily hygiene activities cause harm to the surface of implants. TiN coating enhances hardness, hydrothermal and ozone treatment boosts osteoconductivity and provides decontamination. This activity increases the chance of successful treatment outcome [123]. Efficacy of ozone treatment at implant surfaces also depends on material roughness, since higher roughness increases bacterial adhesion.

In the case of titanium and zirconia, both of which were polished, and zirconia was additionally acid etched, research showed high efficiency of gaseous ozone treatment. The study demonstrated reduction of *P. gingivalis* by more than 99.94% and *S. sanguinis* bymore than 90%, ozone was applied for 24 s at 140 ppm [124]. Isler et al. mention peri-implantitis as a serious problem in dentistry, research demonstrates ozone therapy as an effective adjunction to surgical regenerative treatment. Better results of PI, GI, PD and the amount of bacteria in individual quadrants were observed by scientists. Additionally, no infraction of surfaces, in this case made of titanium and zirconia, was observed [125].

Antimicrobial properties of metal nanoparticles such as titanium dioxide, silver or copper dioxide make it possible to use them as implant modification. Amount of plaque biofilm is an important factor of implantation failure [104]. Titanium alloy is commonly used in dentistry; however, it does not show antimicrobial properties by itself [126].These properties can be improved by applying a surface coating, e.g., visible light active TiO_2._ This material can also be doped with metals and non-metal particles which improve its properties. However, further research is needed to design toxic-free nanoparticles [127]. The TiO_2_ nanoparticle has very beneficial antimicrobial properties which can prevent dental plaque formation [128]. Wettability of surface is important for correct osseointegration but, on the other hand, there are studies that point out better antimicrobial properties at the hydrophobic surface which characterizes TiO_2_ nanotubes. They have good biocompatibility which makes them suitable material for dental implants, they are also suitable for drugdelivery. These properties increase chances of successful treatment [129]. Nano-titania coating also has a positive effect on the peri-implant tissue whose osseointegration process is accelerated [130]. There are many microbes in the oral cavity which is why antibiotics are not effective enough for every pathogen; in addition, bacterial resistance reduces the efficacy of chemotherapy. In TiO_2_ nanotubes can be suspended in various substances, which further enhance its well known microbicidal properties. For better biocompability of implant nanohydroxyapatite, a coating can be added which does not affect the desired properties of titania nanotubes coated with Ag nanoparticles [126,131]. *S. aureus*, *S. epidermidis*, *E. coli* and *P. aeruginosa* are common causes of dental implant complications, such as peri-implantitis. Bacterial infection takes place even 30 min after dental treatment and nanoparticles can help in ensuring correct treatment [132]. Some research demonstrate that nanoporous titania has better antimicrobial properties than nanotube titania material, also the former ensures the release of the loaded drug for 7 days [133]. In Figure 5 has been shown the advantages of using nanoporous TiO_2_ loaded with LL37 which gives positive effect on osseointegration.

Silver NP has a high antimicrobial and osseointegration value which makes this material good for implant coating. It reduces bacterial adhesion up to 50%. The previously mentioned titanium implant has good biocompatibility, which with the addition of silver NP coating, providesan outstanding antimicrobial effect. Nanoporous or nanotube titania coating in combination with silver NP ensure a wide range of antimicrobial properties [133,134]. In addition to the antimicrobial effect which depends on the size of nanoparticle (the smaller it is, the stronger its effect), silver NP coating applied to a titanium implant increases the density of bone [50]. Platinum nanoparticles have a wide range of applications in medicine, such as cancer treatment, bone allograft, bone loss and others. Due to its properties (antimicrobial, anti-inflammatory), PtNP may be suitable for the dental surgeries of guided bone regeneration and guided tissue regeneration [134].

### 3.4. Prosthetic

For removable partial dentures affect, i.e., periodontal tissues, maintaining their hygiene is a very important aspect that affects the long-term use of the prosthetic in the oral cavity by the patient. Therefore, new sterilization methods are needed that help maintain the hygiene of the prosthesis, and thus eliminate the formation of plaque (control its formation). Recently, the use of ozone, which has antibacterial properties, seems to be very promising. The plaque on the prosthesis that causes periodontitis consists mainly of *Lactobacillus*, *Propionibacterium* and *Arcalinia* bacteria. The *Propionibacterium* species were the dominant part of the flora. An equally important factor influencing inflammation of the oral mucosa from the prosthesis are yeasts of the *Candida albicans* genus. *C. albicans* produces large amounts of acetic and pyruvic acid, which causes inflammation. The yeast adheres well to the denture base materials in vitro, although the adhesion depends on the strains and environmental conditions. Denture-induced stomatitis is routinely encountered in clinical practice as a symptom of the plaque build-up on the dental surface (dentures), therefore effective plaque control should be initiated to prevent these consequences. An effective method is the use of ozone as a denture cleaning agent. The advantages of ozone in the water phase are its potency, ease of application, lack of mutagenicity, quick bactericidal action and suitability for use as a solution for soaking medical and dental instruments [134,135]. The use of ozone as a denture cleaner is effective against the methicillin-resistant *S. aureus* and viruses [136]. Ozone can be applied (used) to clean the surface of alloys in removable partial dentures with little effect on alloy quality in terms of reflectance, surface roughness and weight. Direct exposure to ozone gas was a more effective germicide compared to ozonated water. Therefore, ozone gas can be clinically useful for the disinfection of removable dentures [137]. In dentistry, ulcers can occur as a result of using dentures. They are usually located in the vestibular furrows of the maxilla and mandible, causing pain. The treatment of ulcers requires the discontinuation of the use of the prosthesis, its correction, laser therapy and oral and prosthesis hygiene. Treatment also includes the use of topical medications such as chlorhexidine mouthwashes, topical hydrogels, hydrogel dressings, and sometimes topical cortisone. The use of ozone in the treatment of ulcers also seems to be an interesting method due to its properties. Research by Bader et al. on a group of women and men wearing prostheses showed that ozone reduces the size and pain of traumatic ulcer, and accelerates the regeneration of diseased tissue, and thus shortens the treatment [138].

The field of prosthetics is another branch of dentistry in which nanoparticles can be implemented. Their main role is to provide better antimicrobial properties of materials used for making removable dentures. Without proper hygiene, these restorative alternatives can lead to denture stomatitis and cause both reversible and irreversible consequences. One of the studiescompared antifungal activity between a standard denture base (polymethylmethacrylate—PMMA) and one modified with AgNPs. The conclusion was that the addition of AgNPs to the denture base provides 10^5^ less *C. albicans* adhesion than the control group after 24 and 48 h incubation [139]. Another study examined incorporation of silver-sulfadiazine-loaded MSNs into PMMA at up to 5%. Researchers examined mechanical properties and microbial effect against *Candida albicans* and *Streptococcus oralis* on removable and provisional dental restoratives. Results showed that the addition of Ag-MSNs decreased the adhesion of microbes and at the same time improvedmechanical properties of PMMA. The inorganic part of NPs provided better mechanical properties, at the same time, silver ions provided a microbial anti-adhesive effect. Li et al. also examined the addition of silver NPs and their effect on *Candida albicans* adhesion and biofilm formation. Their results demonstrated that the minimal concentration of silver NPs which provides the anti-adhesion effect on *C. albicans* is 5%. Lower concentrations do not ensure this effect [140]. Another study examined the addition of AgNPs to irreversible hydrocolloid impression materials such as Zelgan or Tropicalgin. Results showed better antimicrobial activity compared to standard materials. Adding silver NPs can also provide an increase of gel strength, permanent deformation or the flow of the material, although many of mechanical properties show different variations depending on the wt% of AgNPs [141].AgNPs can also be added to alginate impression powder to increase antimicrobial activity and reduce the risk of cross-contamination by bacteria, viruses or fungi [142]. Köroğlu et al. investigated the effect of adding solution of AgNPs to acryl liquid used for mixing with the powder part of acrylic material. In their study, 0.3, 0.8 and 1.6 wt% of AgNPs were used. Results showed that the addition of 0.8 and 1.6 wt% AgNPs decreased the flexural strength and elastic modulus of microwave-polymerized acrylic resin [143]. In the study of Oei et al., silver ions improved both antimicrobial and mechanical properties of PMMA and the antibacterial activity increased to 28. The study of Matsuura et al. showed that AgNPs implemented in tissue conditioners support antifungal activity against *C. albicans* and antibacterial activity against *S. aureus*, *Pseudomonas aeruginosa* for 4 weeks [144]. Many studies indicate that the addition of AgNPs increases antimicrobial properties of the material and at the same time, doesnotaffect its mechanical properties. It can be considered as a promising component of denture base, dental impression material or tissue conditioner, although further studies are needed. AgNPs can also be considered as a component of permanent prosthetic restoration materials such as porcelain. The idea is to improve fracture toughness and prevention of crack propagation. One of the studies revealed that AgNPs increase the fatigue parameter, shorten the time required for slow crack growth and reduce crack growth rate [145]. Fujieda et al. proved that fracture toughness and Young’s modulus increased with addition of AgNPs and PtNPs [146]. Another study showed that the addition of silver ions improved mechanical properties of Computer Aided Design/Computer Aided Manufacturing (CAD/CAM) blocks and at the same time, decreased crack length [147]. Mohsen et al. examined the effect of AgNPs on the color of ceramics. Their results showed that the addition of AgNPs affected the color quality of dental ceramic. According to numerous studies, AgNPs should be considered as an addition to porcelain, due to improvement of mechanical properties. Another nanoparticle tested in the field of prosthetics is Pt nanoparticles. One of the studies examined modified PMMA denture acrylic containing platinum nanoparticles. The size of the processed Pt NPs was less than 5 nm. Results showed an anti-adherent effect provided by reducing the area available for bacterial adhesion, at the same time, no Pt ion leaching was observed, orit was observed at extremely small amounts [59]. The addition of TiO_2_ nanoparticles in prosthetic treatment has been investigated in many studies. Researchers examined the physical, mechanical and biological behavior of a PMMA/TiO_2_. In one of studies, PMMA/TiO_2_ nanocomposite specimen containing 3 wt. % TiO_2_ NPs was examined. The results revealed that TiO_2_ NPs improved stiffness and mechanical behavior of the PMMA matrix. They also provided both antibacterial and anti-adhesive effects [148]. Some studies suggest that TiO_2_ antimicrobial effect is based on the photocatalytic effect, that is attributable to deactivation of cellular enzymes that lead to higher permeability and to cell death [149]. The study by Tsuji et al. showed that an increased amount of TiO_2_ can weaken the material and cause internal decomposition, therefore adequate amounts of NPs needs to be used [150].

## 4. Cytotoxicity

### 4.1. Ozone

As it was mentioned before, there are three different therapeutic ways in which ozone canbe used in dentistry and other fields of medicine. In dentistry, ozone can be applied in the form of ozonated oil (sunflower oil etc.), ozonated water or simple gaseous ozone.

An in vitro study provided by Borges et al. showed the safety of ozone treatment towards a human skin keratinocyte cell line (HaCaT) and a murine fibroblast cell line (L929). Data demonstrated increased relative cell number of murine fibroblasts and human keratinocytes after treatment with ozone at a concentration of 8 µg/mL.With the use ofa wound healing assay and a scratch assay, it can be clearlyseen that the rate of the wound healing process is the same as in the control group for both cell lines [151]. Another study conducted by Huth et al. evaluated the cytotoxicity of gaseous and aqueous ozone (in PBS) and other antimicrobial agents such as sodium hypochlorite (NaOCl), chlorhexidine digluconate (CHX), hydrogen peroxide 3% (H_2_O_2_) for 1min exposure (a clinically relevant time period for ozonated water and gaseous ozone application) and metronidazole for 24 h exposure to gingival fibroblasts (HGF-1) and human oral epithelial (BHY) cells. Results reveal that less toxic features were observed in HGF-1 and BHY cells treated with ozonated water (1.25 µg/mL), which showed the most biocompatible properties, when compared to gaseous ozone, NaOCl, CHX or H_2_O_2_, which occurred to be the most toxic. An apoptotic assay, used to measure caspase-3 and -7 activity, revealed no alteration after treatment with ozonated water (ozonated PBS) when compared to control in both cell lines. Additionally, metronidazole demonstrated no significant cytotoxic effect towards gingival fibroblasts nor human oral epithelial cells [152]. Similar data was provided later by Colombo et al., who used ozonated olive oil and the CHX agents Corsodyl Dental Gel^®^ and Plak Gel^®^. They evaluated their biocompatibility on immortalized human gingival fibroblasts (HGFs). Each agent was diluted 1:10 four times in Dulbecco’s Modified Eagle’s Medium (DMEM), which resulted in receiving different concentrations of the tested compounds (1:10, 1:10^2^, 1:10^3^, 1:10^4^). After 2 and 24 h of treatment with the aforementioned agents, cells showed 100% viability when treated with ozonated olive oil in all tested concentrations when compared to the control. CHX compounds showed severe cytotoxicity in the highest concentration (1:10) for Corsodyl Dental Gel^®^ and Plak Gel^®^ and moderate cytotoxicity in lower concentrations (1:10^2^, 1:10^3^, 1:10^4^) [153]. Kashiwazaki et al. tested ozonated water and hand disinfectants which contained 83% ethanol, 1% chlorhexidine, 1% chlorhexidine ethanol, 0.2% benzalkonium chloride and 0.5% povidone-iodine on a human keratinocyte cell line in a three-dimensional cultured human epidermis model. Cells were divided into two groups, the first group, 1-week cultured, which developed an immature stratum corneum (SC), and the second group,2-week cultured, which developed a mature SC. Histological changes, cell viability and release of interleukin 1α were evaluated after treatment with ozonated water and hand disinfectant. There was no histological alteration after treatment in 1-week cultured cells and 2-week cultured cells, and also, no vacuolar cell formation was observed. A viability assay showed no cytotoxic effect of ozonated water and interleukin 8 activation was not observed, which indicatesthe high biocompatibility and immuno-compatibility features of the tested ozonated water [154]. Another study evaluated non-toxic and anti-tumor properties of ozonated water in anin vivo mouse model with the useof tumor-bearing mice and the control group. No injurious effect on normal tissue, such as spleen, small intestine, liver, kidney and muscle, was observed after 24 h direct application of ozonated water, even at a relatively high concentration (208 mM). Nonetheless, tumor tissue exhibited a decreased proliferation rate, inhibited growth and showed features of necrosis, which indicatesan increase of ROS levels [155].

### 4.2. Titanium Dioxide

Titanium dioxide nanoparticles (TiO_2_ NPs) can be synthesized via numerous methods such as the sol-gel method, the hydrothermal method, the co-precipitation method, sluggish precipitation, hydrolysis, and simple precipitation [156], but also with the use of the solvothermal or direct oxidation method [157]. With the use of different synthesis methods, we can obtain various sizes of nanoparticles which oscillate from ≤10 to ≥100 nm and even from 200 to 300 nm, and different polymorphic phases such as rutile, brookite or anatase [156,158,159,160].

TiO_2_ NPs are widely used in physics and chemistry due to their photochemical activity and physicochemical stability [158]. Titanium dioxide nanoparticles find their application in gas sensors, solar energy converters, pigments, ceramic supports, but can also be used to purify wastewater [159,161]. Moreover, TiO_2_ NPs are extensively used in UV filters in sunscreens and other everyday cosmeticssuch as lip balm, creams, foundations and also toothpaste [162]. In the food industry, TiO_2_NPs are used as a white pigment in food or food supplements [160]. In dentistry, titanium dioxide nanoparticles can be found in tooth-bleaching gels or dental composites used in orthodontics, in dental acryl resins [163,164].

Due to the many applications of TiO_2_ NPs, it is crucial to evaluate their potential cytotoxic effect on cells and tissues. Skin, gastrointestinal tract and respiratory system are the most exposed to TiO_2_ NPs, especially when applied in cosmetics such as sunscreens, food, or when used in the field of dentistry.

The cytotoxic and genotoxic effect of titanium dioxide nanoparticles was evaluated by Meena et al. The results obtained by them pointed out thedose-dependent (50, 100 and 200 µg/mL) and time-dependent (24, 48 and 72 h) toxic effect of TiO_2_ NPs on human embryonic kidney cells (HEK-293). TiO_2_ nanoparticles induced elevated lactate dehydrogenase releasedfrom the cell, and damage of cell membrane which led to cell death. Moreover, cells treated with 100 and 200 µg/mL of TiO_2_ nanoparticles exhibited an increased level of reactive oxygen species and elevated level of proapoptotic proteins such as Bax, caspase-3 and upregulation of the p53 protein in response to DNA breakage [165]. Another study conducted by Shukla et al. confirmed the cytotoxic and genotoxic effect of nano-TiO_2_ on a human liver cell line (HepG2). Relatively low concentrations of nanoparticles (20, 40, 80 µg/mL) increased the level of ROS, which led to DNA breakage via the oxidative stress-dependent pathway. Additionally, upregulation of p53, Bax, caspase-9 and caspase-3 was observed, however Bcl-2 expression was reduced, which points to apoptosis via the mitochondrial, and thus by the caspase-dependent pathway [166]. This indicates that titanium dioxide nanoparticles should be used with caution and in very low concentrations. Due to the fact that the respiratory system is also exposed to the inhalation of TiO_2_ NPs, it may lead to an increased risk for lung health. TiO_2_ NPs were tested towardsa human lung cancer cell line (A549) to evaluate its potential genotoxic and cytotoxic characteristics. A relatively low concentration of the tested titanium dioxide NPs (10 and 50 µg/mL) after 6to 24 h of incubation led to an increase in the level of ROS, p53 and p21. Similar to the above-mentioned research, Bcl-2 level was downregulated at the mRNA and protein level; moreover, the cleavage of caspase-3 was observed. This data suggests that TiO_2_ NPs cause alteration in gene expression which leads to apoptotic changes and, as a consequence, to human lung cancer cell death [167]. Another research compared five different TiO_2_ NPs: 9 nm rutile (R9), 5 nm rutile (R5) and 14 nm anatase (A14), the commercially available 60 nm anatase (A60) and P25, which contained 80% anatase and 20% rutile, and their potential cytotoxic effect on a human bronchial epithelial cell line (BEAS-2B), a human type II alveolar epithelial cell line (A549) and human bronchial epithelial cells (NHBE). It occurred that the level of ROS was increased especially after 24 h of incubation in all five different TiO_2_ NPs; however, the strongest effect was observed in the P25 sample in NHBE and BEAS-2B cells after 2 and 24 h. For A549 cells, smaller particles (5 and 9 nm) of TiO_2_induced elevated release of intracellular ROS. Cell viability did not drastically alter allofthe tested cell lines, however, after incubation with a relatively high concentration (400 µg/mL), the viability of NHBE and A549 cells slightly dropped. Interleukin 8 (IL-8) plays a key role as a chemoattractant for neutrophils and other granulocytes and stimulates their migration to the infection site. After treatment with five different types of TiO_2_ nanoparticles, the level of the pro-inflammatory mediator IL-8 was substantially elevatedin all the cell lines when compared to the control group. The strongest effect was observed for anatase (A60), however A14 seems not to induce IL-8 upregulation [168]. Titanium dioxide nanoparticles (TiO_2_ NPs) were also tested against a human epidermal cell line (A431) to establish their potential risk to human skin cells. At low concentrations of 8 and 80 µg/mL and short time of incubation (6 h), a decreased level of glutathione and strongly increased level of reactive oxygen species was observed, which led to apoptosis. Genotoxic characteristics, evaluated with the use of a commitment assay, showed damage of DNA at the concentrations of 8 and 80 µg/mL after 6 h of treatment. These data clearly indicate that TiO_2_ nanoparticles should be used at very low concentrations and with high caution [169].

Independent studies conducted by Xue et al., Gao et al. and Wright et al. [170,171,172] have proven the cytotoxic effect of titanium dioxide nanoparticles on the human skin keratinocyte cell line. Xue tested different sizes (4, 10, 21, 25 and 60 nm) and different forms (anatase/rutile, rutile and anatase) under UVA radiation. Results clearly pointed to the toxic effect on HaCaT cells and an increased amount of apoptotic cells, which was induced in a dose-dependent manner, inwhen treated with 10 and 25 nm particles at a concentration of 200 µg/mL. However, UVA radiation alone did not affect cell viability. Elevated levels of reactive oxygen species (ROS) and malondialdehyde (MDA) were also observed and, at the same time, there was a decreased amount of superoxide dismutase, which indicates cell damage [170]. TiO_2_ nanoparticles (25 nm) when compared to nanosized bismuth oxybromide (BiOBr) demonstrated a stronger toxic effect on ahuman keratinocyte cell line. Enhanced ROS was also confirmed at a concentration of 25 µg/mL of TiO_2_NPs. Titanium dioxide nanoparticles also induced early and late apoptosis of HaCaT cells and the cell cycle was found to be arrested after treatment with TiO_2_NPs [171]. Wright analyzed three different sizes of particles (a 1 mm particle composed of 100% rutile, a 21 nm particle composed 80% of anatase and 20% of rutile and a 12 nm particle which contained only anatase). Data demonstrated dose-dependent elevated caspase-8 and caspase-9 activity and increased apoptosis of the tested cells. It is worth noting that cells did not exhibit malignant transformation when treated with TiO_2_ NPs and exposed to UVC radiation [172]. In vivo studies conducted on mice and rat models also confirmed toxic characteristics and increased inflammatory response, which indicates oxidative tissue damage in rats [173]. Mouse model showed in vivo toxicity and DNA disruption of liver tissue and a decrease of glutathione peroxidase, which clearly indicate liver tissue damage [174].

### 4.3. Silver

Many methods are known for the synthesis of silver nanoparticles (AgNPs). With the use of physical, chemical and biological methods, various AgNPs with desirable morphology can be obtained in order to achieve the best properties [175]. Therefore, silver nanoparticles can be used in different fields of science; for example, for single molecule detection [176] or by enhancing electrical conductivity, AgNPs can be intended for high-frequency electronic applications [177]. However, in the biological approach, silver nanoparticles have been known mostly for their antibacterial properties. Due to these characteristics, they can be used as an efficient disinfectant to sterilize surfaces or medical equipment, and devices in the industry during the food packaging process or in environmental usage as air and water disinfectant [178]. AgNPs can also be used to impregnate and functionalize textiles [179], but they can also be used as coating for wood flooring [180].

Despite their various properties, silver nanoparticles exhibit a toxic effect in vitro and in vivo. Many studies revealed that silver nanoparticles increase the level of ROS in cells and therefore increase oxidative stress and cause defects in DNA structure [181,182]. AshaRani et al. [183] evaluated the cytotoxicity effect on human cell lines using human glioblastoma cells (U251) and normal human lung fibroblast cells (IMR-90). The size of the tested Ag nanoparticles oscillated from 6 to 20 nm, however cell viability and ATP level drastically dropped in a time- and concentration-dependent manner, especially after 48 and 72 h incubation. Further investigation revealed mitochondrial damage and increased level of ROS, as well as arrest of the cell cycle in the G2/M phase and disruption of DNA structure [183]. Another study confirmed a dose-dependent and time-dependent toxic effect of silver nanoparticles and silver ions on human osteoblasts (OB) and primary human mesenchymal stem cells (MSC). A severe decrease of cell viability as well as increased oxidative stress was observed already in 10 µg/g of the tested AgNPs after 21 days of incubation with the tested compounds. The human osteoblast (OB) cell line proved to be more sensitive to exposition to silver nanoparticles [184]. Another research confirmed toxic properties of both silver ions and silver nanoparticles towards the human lymphoblastoid TK6 cell line. It occurred that both forms of AgNPs induced cytotoxicity and genotoxicity at comparable concentrations, which oscillated from 1.00 to 1.75 µg/mL of the tested compounds. Additionally, genotoxicity was confirmed with the use of the micronucleus assay and the results clearly showed an increased level of reactive oxygen species (ROS) at relatively low concentrations of both forms of silver after 24 h treatment. The expression of genes involved in oxidative stress, such as glutathione peroxidase 7, thyroid peroxidase and heme oxygenase, and expression of genes in response to cell cycle arrest caused by DNA damage, such as cyclin-dependent kinase inhibitor 1A, were substantially elevated. This study clearly indicates that both silver ions and silver nanoparticles are toxic by leading to an increase in cellular stress and DNA structure damage [185].

The main problem is that in vitro toxicity of silver nanoparticles and silver ions occurs at relatively low concentrations and, on the other hand, antimicrobial properties demand substantially elevated concentrations of AgNPs. Therefore, it is of immense importance to estimate the most favorable conditions, ensuring both antibacterial properties and safe use with regard to cell lines and, even further, for in vivo applications. Hence, Albers et al. [184] estimated in vitro cytotoxicity of silver ions and nanoparticles in osteoclast (OCs) and osteoblast (OBs) cells at antibacterial concentrations against *S. epidermidis*. The study showed a size-dependent and concentration-dependent toxic effect of AgNPs. The most cytotoxic effect was observed after treating OBs and OCs cells with silver nanoparticles sized 50 nm, and the larger the particles were, the weaker cytotoxic effect was observed. This suggests that the smaller the size of the nanoparticle surface is, the more silver is released to the environment. Moreover, OBs cells were more sensitive to the used nanoparticles. As compared with antibacterial activity, which was obtained at a concentration of 8 mg/mL for AgNPs towards *S. epidermidis*, 50% of cell viability was maintained at a concentration of 0.048 mM for osteoclast cells, and for the same concentration, the viability of osteoblast cells was highly below 50%. Thus, at the present moment, there are no possibilities to combine antimicrobial and cyto-safety properties [184]. Pérez-Díaz et al. showed similar data but compared the viability of dermal human fibroblasts at an antimicrobial concentration of sliver nanoparticles against *Streptococcus mutans*. The data clearly indicated that AgNPs concentrations higher than 10 ppm represented acytotoxic level, while reduction of biofilm growth was observed at the concentration of 100 ppm [186].

On the other hand, combined materials, such as Ag NP-coated titanium dental implants with hydroxyapatite (HA) applied to the surface, seem to be promising solutions, because after 7 days, primary human osteoblasts showed biocompatibility with the tested material. Osteoblasts also demonstrated adhesive properties toward the tested material and there were no alterations in cell morphology, and elevated levels of alkaline phosphatase and lactate dehydrogenase were not observed either [187]. This suggests that complex materials such as coating titanium implants with nanoparticles may be the best solution to obtain cyto-safety properties in vivo and in vitro.

### 4.4. Platinum

Platinum, due to its high biocompatibility, its corrosion resistance and the possibility to visualize it under X-rays, is widely used in the field of biomedical sciencesinsurgical instruments, implantable electronic devices and implantssuch as cardioverter-defibrillators, stents, knee or hip implants as well as dental implants [188].

However, platinum nanoparticles (PtNPs) demonstrate different features when compared to platinum solid implants. PtNPs can be obtained via a variety of methods, usually through the reduction of Pt ions in the liquid phase with a stabilizing or capping agent to produce nanoparticles in the form of colloids, eventually via the microemulsion method or via reduction and impregnation of Pt ions into micro-porous base [189]. Due to their method of synthesis, different shapes can be obtained, such as isolated nanoparticles, dendrites or crystalline nanowires with various sizes, which can be used in the industrial application, in optical fields, as a catalyst in fuel cells and biosensors [190,191,192,193].

Some researchers pointed out a size-dependent toxic effect on macrophage cell Raw 264.7 viability. An increased cytotoxic effect was observed when cells were treated with platinum nanoparticles sized 5 nm as compared to PtNPs sized 30 nm, asa smaller size of nanoparticles can be easily taken upby cells; nevertheless, both samples of nanoparticles exhibit a toxic effect. A dose-dependent toxic effect was also observed, cell viability decreased drastically at a concentration of 5 ppm of the tested nanoparticles. Moreover, data indicated that density and cell morphology were altered after treatment with PtNPs. Additionally, platinum nanoparticles induced the activation of caspase-3 and caspase-7, which led to nucleus fragmentation and apoptosis, which also indicates that an elevated level of caspases arrested the DNA repair process [194]. Another study conducted by Konieczny et al. [195] confirmed the cytotoxic effect of platinum nanoparticles on human skin cells. The authors have used PtNPs with an estimated size of around 5.8 and 57 nm, and both were protected with polyvinylpyrrolidone. Nanoparticles were used with three different concentrations of 6.25, 12.5 and 25 µg/mL and, after 24 and 48 h treatment, the viability of normal human epidermal keratinocytes (NheKs) slightly decreased, especially when treated with smaller particles. Nonetheless, cell viability seems to be unaffected, and genotoxicity and DNA damage via activation of caspase-3 and caspase-7 were observed, primarily caused by smaller PtNPs. This study indicated dose-dependent and concentration-dependent toxic properties of platinum nanoparticles [195]. In Labrador-Rached et al.’s research, it was demonstrated that 70 nm PtNPs in a concentration of 100 µg/mL exhibit a relatively high cytotoxic effect, however a toxic effect observed at lower concentrations of 5 and 25 µg/mL was indiscernible. They also obtained elevated ROS levels in response to an upregulated release of pro-inflammatory factors such as IL-1β, IL-8 and TNF-α in a HepG2 liver model [196].

An in vivo model provided by Lin et al. showed that small platinum nanoparticles of 5 nm and larger, up to 70 nm, by acting on ion channels by the extracellular site of ventricular cardiomyocytes of neonatal mice, disrupted cardiac electrophysiology, which can lead to the threat of cardiac conduction block. However, a relatively high dose of 5 and 70 nm PtNPs did not meaningfully elevate ROS generation and lactate dehydrogenase level [197].

To compare toxic nanoparticles, green synthesis of platinum nanoparticles could be a good solution to reduce their toxic properties [198]. Some studies still demonstrate toxicity and genotoxic effect on various cell lines, such as human embryonic kidney (HEK293) cells [199] or human breast cancer (MCF-7) cells [200]. However, novel studies reveal a selective cytotoxic effect on cancer cells (MCF-7) when compared to normal human embryonic kidney cells (HEK293) [201], which could be a good direction for the usage of this particles.

## 5. Discussion

Ozone has a wide range of applications. In dentistry, its properties are not only antimicrobial, but immuno-stimulating, analgesic and anti-inflammatory. The use of ozone in dental surgery seems promising, especially because of its wound-healing properties. There are many caries prevention agents on the market: ozone seems to be effective against microbes after cavity preparation, but the opinions of scientists are divided [71,72]. The problem in the case of ozone is its efficacy against biofilm. Ozone therapy can help maximize disinfection effect, at the same time providing low toxicity and low possibility of drug resistance. Although ozone effectiveness shows a wide range in many studies, it can be considered as an additional disinfection protocol step. New technologies in dentistry are most often associated with a high cost of their introduction to the dental office. What also poses a problem in the case of ozone generators are substitutes that do not require investment and are at the same time effective, examples are fluoride varnish and sealants for caries prevention or NaOCl used in endodontics [73]. Apart from the discussed ozone applications in dentistry, there are many other fields in medicine where ozone can have a significant impact. Acceleration of wound-healing of oral mucosa is an example of positive properties, which offers a wide range of application possibilities [202].

Ag NPs are the most frequently discussed NPsin the dentistry literature. Due to their properties, they are widely used in medicine, for example in the form of coating in dental implants or as a modifier of dental materials. Also, many studies suggest that AgNPs can be considered in endodontic protocols as an additional irrigation or amplifier of the antimicrobial effect of other agents, such as MTA and CHX. However, the problem is to find the most favorable ratio between cytotoxicity and antimicrobial properties, which depends on the size of NPs. TiO_2_ are used especially as coatings of implants or rotary NiTi instruments. They can be combined with other NPs such as Ag or ZrO_2_NPs to obtain better properties of materials. Nanoporous and nanotube titania coating may contain various substances, such as antibiotics, silver and zirconia, which improve their properties [126]. TiO_2_ NPs appear to have a promising reduction effect on biofilm which is a serious problem in dentistry. However, the problem of cytotoxicity arises again. Silver and titania NPs appear more often in literature and their antimicrobial properties seem to be well proven in contrast to CuO and Pt NPs, which do notappear frequently in articles. Pt NPs have promising properties, especially in guided bone and tissue regeneration. Also, in the field of prosthetics, theyare used in modified PMMA denture acrylic. However, again the problem is their cytotoxicity and the possibility of using other NPs [203]. CuO nanoparticles are used in conservative dentistry as a caries preventing agent and they seem to have a potential for application in endodontic materials. Unfortunately, cytotoxicity is a significant problem.

Different antimicrobial agents are reported to be used as the components of orthodontic adhesives. Among them are metal nanoparticles which, regarding the growing bacterial resistance to the commonly used drugs, may be considered as an alternative treatment applied as a peri-implant infection prevention measure [68]. Generally, nanoparticles, due to their small size as well as large surface area, possess unique features and, especially when combined with metal ions, can help prevent bacterial growth in the vicinity of the material [68]. The efficacy of antimicrobial treatment with inorganic nanoparticles is highly correlated with their physicochemical properties. In general, the smaller size of grains the material has, the more effective it is. Another important factor, apart from the size of particles, is their morphology, and it was established that needle-shaped particles are more likely to damage bacterial cells than spherical ones [204]. Metal nanoparticles may have bactericidal or bacteriostatic effects. The result of the former is bacterial death and the latter leads to the inhibition of growth or multiplication [204]. Most frequently, the antimicrobial effect of applied nanoparticles is based on the release of free metal ions, although other modes of action have also been noted, such as direct mechanical damage of bacterial cell after the internalization of the particles into the cell. Another mechanism is connected with the production of reactive oxygen species (ROS) [204]. Nanoparticles can also adsorb to the cell wall and, as a result of depolarization, increaseits permeability [205]. Moreover, metals lead to protein dysfunction and impair the enzymatic activity, which results in cellular metabolism malfunctioning [206]. Nanotechnology provides a wide range of modifications in dentistry, where new, better materials or implants are constantly being sought. The addition of nanoparticles can improve their properties such as their bactericidal, adhesion, or osseointegration properties. The discussed materials have a variety of applications in many fields of dentistry. The issue which is mainly elaborated in this reviewisantimicrobial properties. However, nanomaterials offer many more possibilities, as they have even more properties which enhance materials and tools used in medicine.

## 6. Conclusions

In summary, both ozone treatment and nanotechnology seem to have a prosperous future in dentistry, offering a wide range of applications. Ozone can be used in every field of dentistry due to its efficient antibacterial properties. Treatment with ozone may be more appropriate for people with visit anxiety. The role of Ag and TiO_2_ NPs in dentistry seems to be promising. In the case of CuO and Pt NPs, the problem may be the cytotoxicity and the presence of alternative materials, e.g., other NPs. It is important to compare the ratio of the effectiveness of the discussed agents to those commonly used. Their price may also pose a problem, especially in the case of prophylactic treatments, such asozonehygienization and povidone iodine impregnation.Both ozone and nanoparticles discussed in this review have antimicrobial properties. For this reason, they can be used as an improvement to treatment methods or materials. Another advantage is the prevention of complications such as peri-implantitis and secondary caries. Nanotechnology gives a huge field for the development of new materials and methods of treatment in modern dentistry. Further research is needed for each of the agents to rediscover or find the most advantageous method of obtaining the material or using it in dentistry.

## Figures and Tables

**Figure 1 nanomaterials-11-00259-f001:**
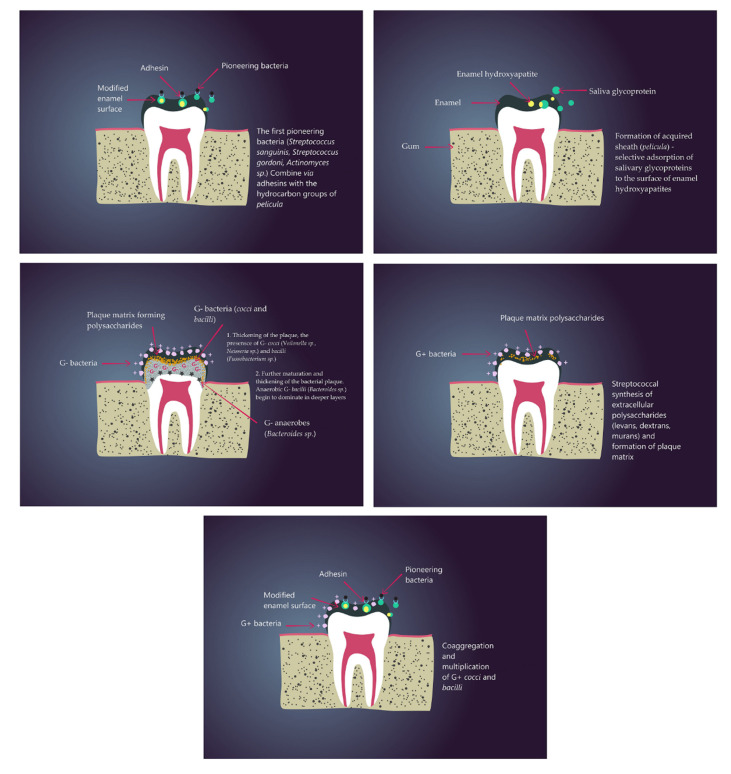
A scheme showing the stages of biofilm formation.

**Figure 2 nanomaterials-11-00259-f002:**
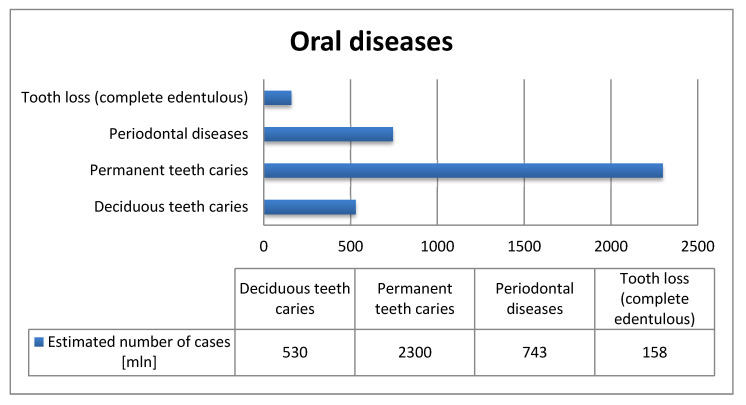
A graph showing the number of patients affected by oral diseases. Teeth caries is one of the most common illnesses in the world.

**Figure 3 nanomaterials-11-00259-f003:**
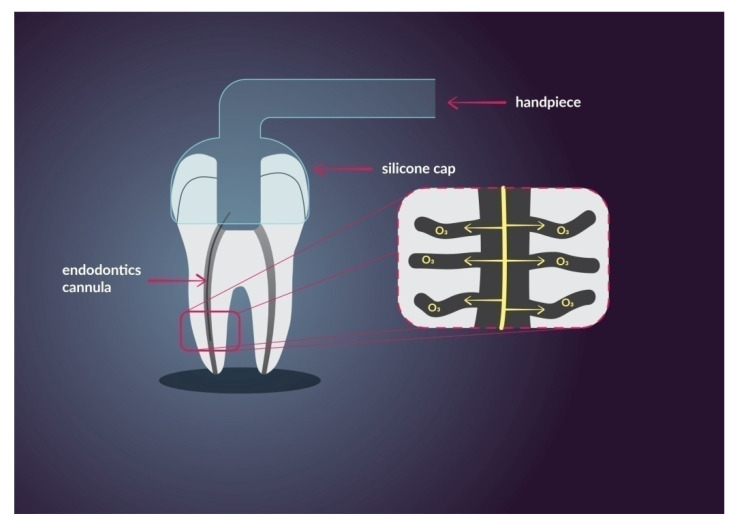
Aschemeof application of ozone treatment.

**Figure 4 nanomaterials-11-00259-f004:**
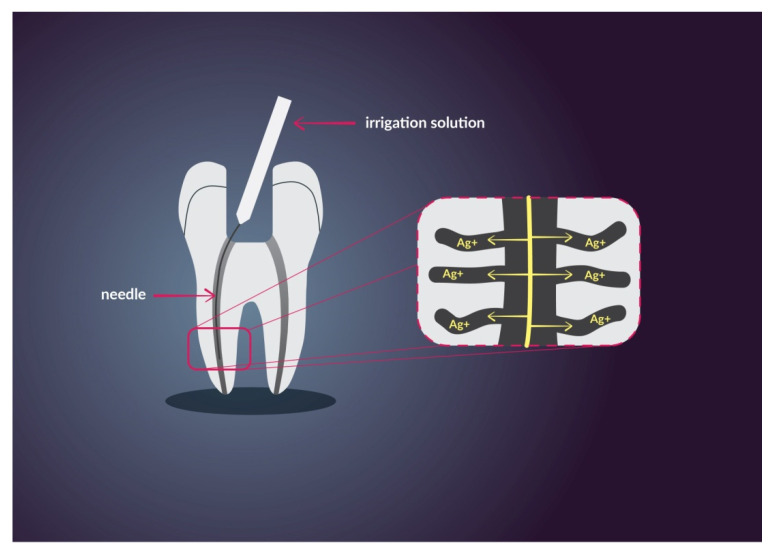
A scheme of irrigation with AgNPs solution. AgNPsreleaseAg^+^ions.

**Figure 5 nanomaterials-11-00259-f005:**
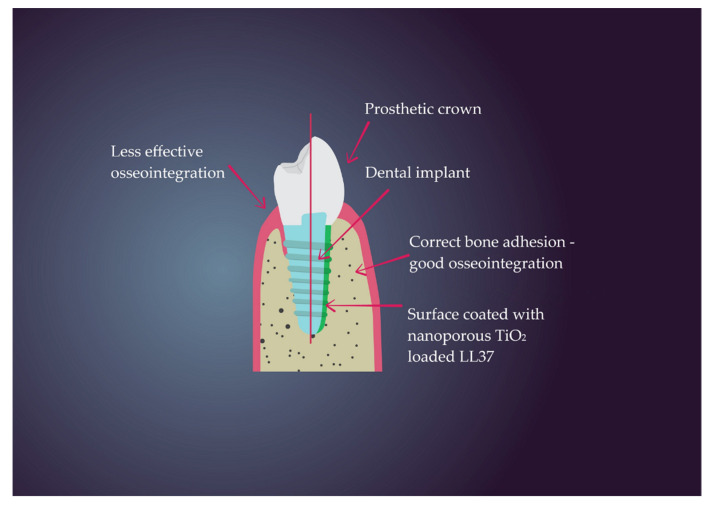
Ascheme showing difference in osseointegration between implant surface with and without coating.

## Data Availability

Not applicable.

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
