# Peer review of "Application of Selected Nanomaterials and Ozone in Modern Clinical Dentistry"

_nanomaterials, 2021, doi:10.3390/nano11020259_

Round 1
Reviewer 1 Report
Review appears complete and well focused. However, conclusions should be more properly written.
Minor points need to be addressed:
- Please note that on page 3 lines from 88 to 99 contain journal instructions
- Line 80 “100nm” needs space
- Authors go from section 1 directly to section 3
- Line 120 “measureg” should be corrected
- Line 128 “aforementioned” should be corrected
- Line 135 “1- week” has extra space
- Line 147 “ROS level” should be corrected in levels
- Line 162 “Streptococcus” should be indicated in Italics
- Line 196 “via” should be indicated as Italics
- Line 210 comma is missing
- Line 222 “At” should be removed
- Line 223 “they” should be removed
- Authors should select between “hours” or “h”
- Line 255 “uses” slould be substituted with “used”
- Line 275 extra space is present
- Line 305 comma is missing
- Line 322 comma is missing
- Line 362 concentration value is missing
- Line 364 “a” should be substituted with “an”
- Line 382 extra comma is present
- Line 386 “i.a.” should be deleted
- Line 403 50 should appear as subhead
- Line 406 comma is needed
- Line 433 extra space is present
- Line 436 via should be indicated as Italics
- Line 438 via should be indicated as Italics
- Line 457 via should be indicated as Italics
- Line 470 “For comparatively” should be substituted by “To compare”
- Line 478 via should be indicated as Italics
- Line 480 via should be indicated as Italics
- Line 498 via should be indicated as Italics
- Line 507 via should be indicated as Italics
- Line 512 spaces are missing
- Line 525 50 should appear as subhead
- Line 530 space is missing
- Line 535 50 should appear as subhead
- Line 548 space is missing
- Line 558 has a problem
- Legend of Figure 2 should be justified
- Line 571 space is missing
- Line 578 one “the” should be deleted
- Line 617 S. mutants should be indicated as Italics
- Line 623 Candida should be indicated as Italics
- Line 624 Candida should be indicated as Italics
- Line 665 point is missing
- Line 681 “usge” should be substituted by “usage”
- Line 692 one “that” should be deleted
- Line 700 space is missing
- Line 740 space is missing
- Line 754 space is missing
- Line 773 demonstrate should be corrected in demonstrates
- Line 787 this should be substituted by these
- Lines 791-794 the sentence appears wrong
- Lines 809-810 commas are missing
- Line 855 improved should be correctly written
- Line 859 “og” should be substituted by “of”
- Line 896 attribuotable should be corrected
- Line 908 “of” is missing
- Line 935 “seems” should be corrected with “seem”
- Line 946 space is missing
- Line 954 which should be corrected
- Line 966 2 should be subhead
Author Response
Dear Editor,
We would like to express our sincerest gratitude to the Reviewers for their enormous efforts in criticizing the manuscript. We have taken into account all raised question here follows the detailed answers to the Reviewers. Moreover, all changes we have made to the original manuscript, are marked in the red colour in the text.
Reviewer 1
Review appears complete and well-focused. However, conclusions should be more properly written.
ANSWER: Correction has been made.

Reviewer 2 Report
This review paper is focused on reviewing the state of the art of the utilization of ozone and select nanoparticles in dentistry applications. This review is very well written and provides an in-depth literature review of this important topic. However, there are a few modifications that should be addressed prior to publication.
1) The whole article could stand to have better organization to improve readability. Right now, there are full page paragraphs, which can easily lose reader interest. It is suggested that additional sub-categories and topics be included to improve organization and flow.
2) Section 2 is missing
3) There is not a clear understanding of why this review paper is looking into both ozone and multiple nanoparticles. Please provide additional introduction that would connect these two in dental applications.
4) I would suggest removing copper oxide. Due to the severe toxic concerns, these particles are not being used in modern dentistry or medical applications anymore.
5) Additional overview figures would improve this review.
6) Please consider moving section 4 up in the review article. Placing the current dentistry applications prior to the toxicity sets the stage for why the toxicity is of concern in this review.
Author Response
Dear Editor,
We would like to express our sincerest gratitude to the Reviewers for their enormous efforts in criticizing the manuscript. We have taken into account all raised question here follows the detailed answers to the Reviewers. Moreover, all changes we have made to the original manuscript, are marked in the red colour in the text.
Reviewer 2
This review paper is focused on reviewing the state of the art of the utilization of ozone and select nanoparticles in dentistry applications. This review is very well written and provides an in-depth literature review of this important topic. However, there are a few modifications that should be addressed prior to publication.
1) The whole article could stand to have better organization to improve readability. Right now, there are full page paragraphs, which can easily lose reader interest. It is suggested that additional sub-categories and topics be included to improve organization and flow.
ANSWER: Section 3 has been split into section 2 (due to incorrect numbering) and section 4 that should improve the readability.
2) Section 2 is missing.
ANSWER: Numbering has been corrected.
3) There is not a clear understanding of why this review paper is looking into both ozone and multiple nanoparticles. Please provide additional introduction that would connect these two in dental applications.
ANSWER: This review has focused on a use of nanotechnology and ozone molecules in modern dentistry. The antibacterial properties of these agents can allow more effective methods of treatment with fewer complications.
4) I would suggest removing copper oxide. Due to the severe toxic concerns, these particles are not being used in modern dentistry or medical applications anymore.
ANSWER: I do not agree with Reviewer to removing the part related to copper oxide. Of course, it is a truth that in modern dentistry a use of the copper oxide is limited but there is a new trend related the nanosized cooper oxide to be applied in nanomedicine [[1]].
5) Additional overview figures would improve this review.
ANSWER: The Figures have been improved.
6) Please consider moving section 4 up in the review article. Placing the current dentistry applications prior to the toxicity sets the stage for why the toxicity is of concern in this review.
ANSWER: Section 4 now concern toxicity (Numbering has been changed due to earlier lack of section 2)
Reference:
[1] F. A. Bezza, S. M. Tichapondwa and E. M. N. Chirwa, Scientific Reports, 10 (2020) 16680
